

# Cloning and functional characterization of porcine *AACS* revealing the regulative roles for fat deposition in pigs

Pan Zhang[1,2], Bo Zhang[1], Yu Fu[1], Pan Li[1] and Hao Zhang[1]

[1] State Key Laboratory of Animal Biotech Breeding, China Agricultural University, Beijing, China
[2] Beijing Academy of Science and Technology, Beijing Milu Ecological Research Center, Beijing, China

## ABSTRACT

Fat deposition is a quantitative trait controlled by multiple genes in pigs. Using transcriptome sequencing, we previously reported that *AACS* is differentially expressed in the subcutaneous fat tissue of Dingyuan pigs with divergent backfat thickness. Therefore, with the aim of further characterizing this gene and its protein, we cloned the entire 3286-bp mRNA sequence of the porcine *AACS*, and the encoded AACS protein is a hydrophilic protein without a signal peptide or transmembrane sequence. Our findings suggested that among various tissues and pig breeds, *AACS* was highly expressed in subcutaneous fat. We have identified three completely linked SNP loci in the AACS gene: A-1759C, C-1683T, and A-1664G. The double luciferase activity test in the 5′ flanking region indicated that the flanking region of *AACS* contained several active regulatory elements. The three linked SNPs that were identified in one of the critical active elements, and might serve as important molecular markers regulating backfat thickness. Finally, we observed that *AACS* overexpression inhibited the proliferation and differentiation of subcutaneous preadipocytes. Collectively, our results suggest that *AACS* inhibits subcutaneous fat deposition in pigs. This study provides a new molecular marker for understanding the mechanism of porcine fat deposition.

## INTRODUCTION

The main adipose tissue depot in pigs is the subcutaneous adipose tissue (SAT), accounting for over 70 % of the total body fat in pigs (*Allen, 1976*). In the actual production process, the excessive accumulation of subcutaneous fat increases production costs and reduces product quality, thus affecting economic benefits. Subcutaneous fat reduction is a critical aspect of pig breeding (*Hocquette et al., 2010*; *Louveau et al., 2016*). Interpreting the genetic determinants of fat deposition and identifying molecular markers that influence fat deposition are the theoretical basis for conducting molecular breeding; therefore, candidate genes that control subcutaneous fat deposition must be investigated.

Acetoacetyl-CoA synthase, also known as acetylacetic acid CoA (*AACS*), is a cytoplasmic ketone body (acetylacetic acid)-specific ligase found in various adipogenic tissues (*Ito et al., 1986*) that specifically converts acetylacetic acid to acetoacetyl-CoA for the synthesis of cholesterol and fatty acids (*Endemann et al., 1982*) . *AACS* mRNA is especially abundant in

Corresponding author
Hao Zhang, zhanghao827@163.com

white adipose tissue, and its expression increases during adipocyte differentiation (*Yamasaki et al., 2007*). The expression pattern of *AACS* during preadipocyte differentiation is highly similar to that of acetyl-CoA carboxylase 1 (ACC-1), a key enzyme in fatty acid synthesis, suggesting that the function of *AACS* is related to fatty acid synthesis (*Yamasaki et al., 2007*; *Yamasaki et al., 2005*; *Yamasaki et al., 2009*). The C/EBP-$\alpha$ binding site was found in the *AACS* promoter region, suggesting that (C/EBP-$\alpha$) is crucial for *AACS* expression in adipocyte differentiation (*Hasegawa et al., 2008*). *AACS* plays a vital role in the regulation of fat deposition.

We previously identified *AACS* as the key differentially expressed gene in the subcutaneous fat tissue of Dingyuan (DY) pigs with divergent backfat thicknesses using transcriptomic profiles (*Zhang et al., 2022b*). Therefore, we surmised that *AACS* is a crucial candidate gene for back subcutaneous fat thickness in pigs. Accordingly, we cloned porcine *AACS* to evaluate its role in regulating fat deposition and identified its functional molecular markers that might regulate its transcriptional expression. We believe that our results would help identify major genes or markers that may benefit the pig-breeding industry.

## MATERIALS & METHODS

### Ethics statements

Animal rearing and handling were performed in accordance with the Guide for the Care and Use of Laboratory Animals of China. All experimental protocols were approved by the Committee on the Ethics of Animal Experiments of the China Agricultural University (permit number: SKLAB-2012-04-07).

### Experimental materials

Ear tissue samples were collected from Tibetan pigs (TP, $n = 34$), Landrace (LL, $n = 34$), Yorkshire (YY, $n = 40$), and Berding (a population that was bred from hybrid breed of Berkshire pig and Dingyuan pig, BD, $n = 104$) pigs for DNA extraction. The TP, LL, and YY pigs were raised at the Tibet Agriculture and Animal Husbandry College, while BD and DY pigs were raised at the Ankang Agriculture and Animal Husbandry Company, Dingyuan County, Anhui Province. BD individuals had phenotypic data of backfat thickness and age up to 70 kg. RNA was extracted from the subcutaneous fat tissue and longissimus dorsi muscle of six-month-old TP ($n = 6$), YY ($n = 6$), and DY ($n = 6$). Based on our observation and previous studies, the TP is a high backfat thickness breed, while the LL and YY pigs are low backfat thickness breeds (*Zhang et al., 2022a*; *Wang et al., 2014*). In our previous study, the DY pigs with high backfat thickness (HBF, $n = 6$) and low backfat thickness (LBF, $n = 6$) groups were selected from a DY population at similar age old and body weight (*Zhang et al., 2022b*). Six-day-old piglets were used to isolate primary preadipocytes. The animals are raised under the same conditions and have free access to food and water. The feed provided to the animals has a metabolizable energy content of 13.5 MJ/kg and a crude protein content of 16.0%. The pig ear samples were obtained as a general breeding monitoring procedure, after which the pigs were continued to be fed at their original breeding base. The animals that were used to collect the back adipose tissue for RNA extraction and cell isolation were slaughtered in accordance with the standard

"Operating procedures of livestock and poultry slaughtering—Pig" (GB/T 17236-2019, China). Briefly, the pigs were fasted for 12 h with free access to fresh water. After that, the pigs were shocked at 90 V and 50 Hz for 10 s to stun them and then exsanguinate.

### DNA, RNA, and cDNA preparation

Genomic DNA was isolated from the ear tissue using a standard phenol/chloroform extraction method. Total RNA was extracted from tissue samples with TRIZOL® Reagent (Invitrogen, Carlsbad, CA, USA). Use a NanoDrop 2000 Biophotometer (Thermo Fisher Scientific, Waltham, MA, USA) and electrophoresis to verify its quality and integrity. Following the manufacturer's instructions, we used the FastQuant Reverse Transcription Kit (TIANGEN, Beijing, China) to reverse transcribe 2 μg of RNA sample in a 20 μL reaction volume into cDNA.

### Cloning and sequence analysis of AACS

The 5′- and 3′-end sequences of the cDNA encoding the *AACS* were obtained from the adipose tissue of TP using the Smarter™ RACE cDNA Amplification kit 5′/3′ according to the manufacturer's instructions, and the remaining sequences were amplified using PCR; the specific primer sequences are listed in Table S1. Gel purification of PCR products, ligation to vectors, splicing of full-length mRNA sequences of porcine AACS genes, and prediction of physicochemical properties of AACS protein (including amino acid hydrophobicity, theoretical isoelectric point, and instability point index) were performed according to the methods described previously (*Zhang et al., 2022a*). Online software Signal P 3.0 (http://www.cbs.dtu.dk/services/Signal/P-3.0/) and TMHMM (HTTP://www.cbs.dtu.dk/services/TMHMM) were used to predict the signal peptide region and transmembrane structure of the protein, respectively. The BLast tool (Basic Local Alignment Search Tool, https://blast.ncbi.nlm.nih.gov/Blast.cgi) from NCBl was used to perform a comparison of amino acid and nucleotide sequence similarity.

### Measurement of gene expression

A semi-quantitative reverse transcription and polymerase chain reaction (SqRT-PCR) was used to detect *AACS* expression in several pig tissues, and the assay was performed as previously described (*Wang et al., 2014*). Total RNA (2 μg) was reverse transcribed to cDNA using the FastQuant RT Kit (TIANGEN, Beijing, China) following the manufacturer's specifications. Then, quantitative real-time PCR (qPCR) was performed using SYBR Green qPCR SuperMix (TIANGEN, Beijing, China). The amplification cycle system was as follows: 95 °C for 15 min, followed by 40 cycles of 95 °C for 15 s, 60 °C for 30 s and 72 °C for 30 s. The relative gene expression levels were calculated using the $2^{-\Delta\Delta Ct}$ method. The specific primer sequences are presented in Table S2, and the housekeeping gene, $\beta$-actin, served as the positive control.

### SNP screening, genotyping, and correlation analysis

To detect potential functional regulatory sites in the 5′ lateral region, three pairs of primers for SNP screening of the *AACS* were designed using Primer Premier software (version 5.0; Premier Biosoft, Palo Alto, CA, USA) and are listed in Table S3. The amplicon sequences

covered 2002-bp regions in the 5′-flanking (numbered starting from the start codon; the first base upstream of ATG was designated as −1, followed by sequential numbering until −2,164). First, we performed pooled sequencing using 8 individuals from each breed (TP, YY, and LL) to identify the existendce of SNPs. For sites with SNP, we amplified the DNA from each individual (TP, $n = 34$; LL, $n = 34$; YY, $n = 40$) separately for gene genotyping analysis. A single bright band of the final product was considered as qualified and sent to SinoGenoMax for sequencing. Chromas Pro (Technelysium Pty) and DNAMAN (version 8.0) were used to analyze sequence variation. Chi-square ($\chi^2$) test of SPSS (version 21.0) was used to analyze genotype distributions and differences.

## Dual-luciferase reporter assays

To detect the *AACS* gene promoter region activity, the −2044–+116 fragment of this promoter region was amplified and then cloned into a pGL3-Basic vector using homologous recombination. The primers used for amplification are listed in Table S4. The target plasmid (900 ng) was co-transfected with the internal control plasmid pRL-TK (100 ng), while pRL-TK was co-transfected with empty pGL3-Basic as the external control. There were 4 replicates in each group. At 48 h post-transfection, the cells were lysed in 100 µL of lysis buffer and then assayed for promoter activity using a dual-luciferase reporter assay system (Promega, Madison, WI, USA) on the PerkinElmer 2030 Multilabel Reader (PerkinElmer, Waltham, MA, USA). The Renilla luciferase signal was normalized to the firefly luciferase signal.

## Isolation and culture of subcutaneous preadipocytes

Subcutaneous adipocytes were stripped from subcutaneous deposits in the neck and back of six-day-old piglets. The adipose tissue was cut into small pieces and digested with 1 mg/mL collagenase type I (Invitrogen, Carlsbad, CA, USA) at 37 °C for 60 min in a reciprocating shaker bath, followed by filtration. Preadipocytes were cultured in a growth medium (GM) containing Dulbecco's modified Eagle medium (DMEM; Gibco, Grand Island, NY, USA) supplemented with 10% heat-inactivated fetal bovine serum (FBS; Gibco) and 1% penicillin/streptomycin (PS, Gibco). All cells were maintained in a humidified atmosphere containing 5% $CO_2$ at 37 °C.

## Induced differentiation of porcine preadipocytes

When the primary cells reached 100% confluence, adipogenesis was induced with a differentiation medium for 2 days, which consisted of DMEM supplemented with 10% FBS, 0.5 mM isobutylmethylxanthine, 20 nM insulin, and 0.5 mM dexamethasone. Subsequently, cells were cultured in a maintenance medium consisting of 5 µg/mL insulin in GM.

## Plasmid construction, lentivirus packaging, and lentivirus infection

To amplify the CDS region of the porcine *AACS*, the corresponding fragment was inserted into the EcoRI/BamHI site of the pLenti-CMV-EGFP-3FLAG-PGK-Puro vector (Obio Technology, Shanghai, China). Using Lipofectamine 2000-mediated transfection, 293T cells were co-transfected with the pLenti-CMV-EGFP-3FLAG-PGK-Puro vector, vesicular

stomatitis virus G protein plasmid, or packaging plasmid. After 48 h, the cells generated mature lentivirus-containing supernatant. The preadipocytes were infected with the lentivirus and then screened with a concentration of 5 μg/mL of puromycin to obtain a stable *AACS*-overexpressed preadipocytes line.

### Oil Red O staining and dye extraction analysis

After removing the growth medium, cells were fixed with 4% paraformaldehyde for 30 min at room temperature. The cells were washed with PBS and stained with Oil Red O working solution (Sigma, St. Louis, MO, USA) for 30 min. The cells were again washed with PBS and observed under a microscope. Finally, Oil Red O dye was extracted from the stained cells with isopropanol for 20 min, and the lipid droplet content was evaluated by spectrophotometrically measuring the absorbance at 490 nm. There were 3 replicates in each group.

### CCK8 and EdU proliferation assays

The cells were inoculated in 96-well plates, and 10 μL of CCK8 (Beyotime Biotechnology, Shanghai, China) solution was added to each well before the assay. There were 9 replicates in each group. After incubation for 1 h, the absorbance was measured at 450 nm using a microplate reader (Biotek, Winooski, VT, USA).

The EdU assay was performed using an EdU assay kit (Beyotime, Shanghai, China) according to the manufacturer's instructions. Cell nuclei and EdU-positive cells were stained blue and red, respectively ($n = 3$).

### Statistical analysis

Statistically significant differences were determined *via* the *t*-test using SPSS software (version 21.0; IBM, Chicago, IL, USA). Graphs were prepared using GraphPad Prism 7 software (GraphPad Software, San Diego, CA, USA). The results are expressed as mean ± standard deviation (SD), with statistical significance set at $p < 0.05$ and highly significant values at $p < 0.01$.

## RESULTS

### Cloning and sequence analysis of *AACS*

The complete mRNA sequence of *AACS* was successfully cloned using 5′ rapid amplification of cDNA ends (RACE), 3′ RACE, and PCR (Fig. 1A). The sequence with a 5′ end length of 1892 bp was obtained using nested PCR, with the first amplified product being 2211 bp. The sequence with a 3′ end length of 577 bp was obtained using 3′ RACE, and the intermediate sequence was obtained using PCR amplification with 1575 bp. After sequencing and assembling, the full-length 3286-bp *AACS* sequence (Accession Number, OP807955), including the coding sequence of 2019 bp, 5′ untranslated region of 161 bp, and 3′ untranslated region of 1106 bp, encoding a total of 672 amino acids, was obtained. Sequence analysis with ExPASy revealed that the isoelectric point of the AACS protein was 5.79, with a relative molecular mass of 75 kDa and an instability index of 32.82; therefore, it was presumed to be a stable protein. Hydrophobicity prediction analysis revealed strong
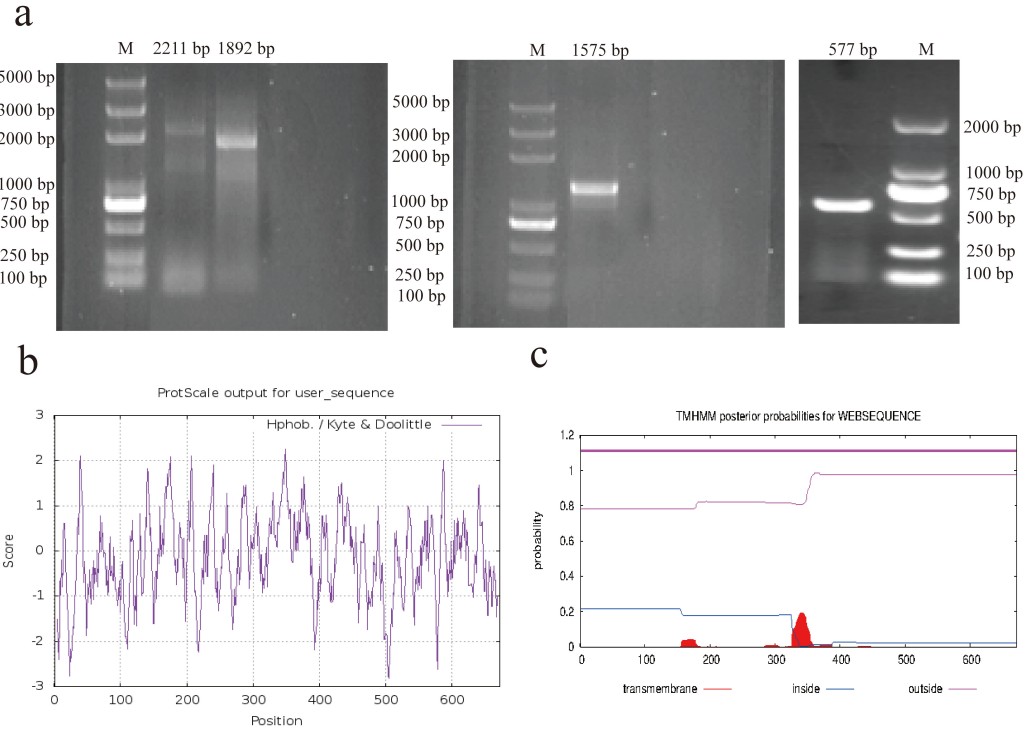

**Figure 1** **RACE amplification and sequence analysis.** (A) Amplification of 5′ RACE (left), PCR (middle), and 3′ RACE (right). (B) Prediction of hydrophilicity of the AACS protein. (C) Transmembrane domain prediction of the AACS protein. M = Trans2K/Trans2K Plus DNA Marker(Transgene, Beijing, China).

hydrophilic regions in many parts of the protein. The hydrophobic regions were evenly distributed with an average value of −0.170, indicating that *AACS* is hydrophilic (Fig. 1B). Furthermore, no signal peptide sequence or transmembrane domain was present in this gene (Fig. 1C). The results indicated nucleotide similarities of 89.71, 89.14, 88.44, 88.15, and 86.68% and amino acid sequence similarities of 93.9, 93.45, 91.82, 91.67, and 90.62% for cattle, sheep, donkeys, horses, and humans, respectively, suggesting that the *AACS* is relatively conserved among these species.

## Expression profile of porcine AACS in tissues

The *AACS* expression in the lung, liver, kidney, leg, hypothalamus, back fat (BF), heart, and longest dorsal (LD) muscle tissues of pigs were evaluated. The results revealed that *AACS* was expressed in all eight tissues, with high expression in the heart and BF tissues (Fig. 2A). To investigate *AACS* function in fat deposition, *AACS* expression was examined in the BF and LD tissues. In BF tissue, the mRNA level of *AACS* in TP was significantly lower than that in YY, and no significant differences were observed between these in LD tissue (Fig. 2B). Similar results were obtained in DY pigs with divergent backfat thicknesses. In BF tissue, *AACS* expression was significantly lower in the HBF group than in the LBF group,
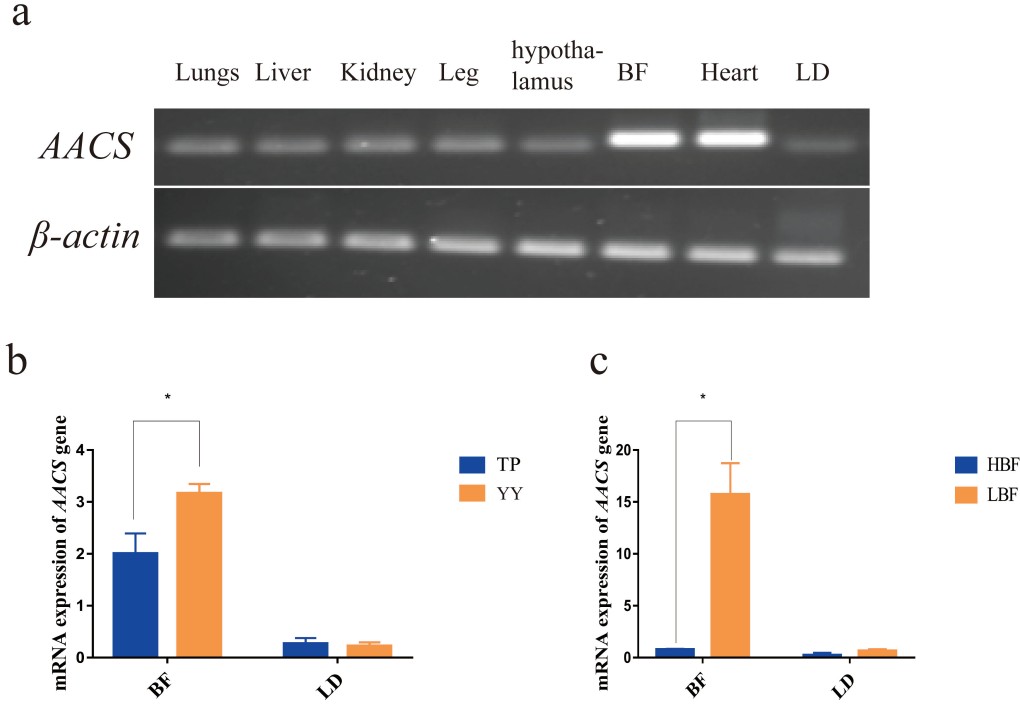

**Figure 2  Expression analysis of *AACS* in tissues.** (A) *AACS* expression in different tissues of TP pigs using SqRT-PCR. (B) The mRNA expression of *AACS* in TP and YY pigs. (C) The mRNA expression of *AACS* in DY pigs. BF, back fat; LD, longest dorsal; TP, Tibetan pig; YY, Yorkshire pig; HBF, high back fat thickness; LBF, low back fat thickness. Each bar represents the means ± SD. * $P < 0.05$.

with no significant difference in LD tissue (Fig. 2C). These results suggest that *AACS* might play a negative regulatory role in BF deposition in pigs.

## SNP identification of 5′ flanking region in *AACS*

Using Sanger sequencing, three complete linkage mutation loci in the 5′ flanking region, namely A-1759C, C-1683T, and A-1664G, were identified (Fig. 3). The three SNPs were completely linked and formed two haplotypes, ACA and CTG. Thus, the frequencies of allele and genotypes of A-1759C could represent the polymorphisms at these three sites. For site A-1759C, three genotypes (AA, AC, and CC) were observed in the LL and YY groups, and two genotypes (AA and AC) in the TP group, and the genotype frequency distribution was significantly different between the two lean-type pigs (LL and YY) and the TP group (Table 1). This linkage site is likely a crucial SNP marker associated with fat deposition.

To confirm this hypothesis, we genotyped A-1759C in 104 BD pigs and analyzed its association with backfat thickness and age up to 70 kg. The results helped identify 34, 58, and 12 pigs with the AA, AC, and CC genotypes, respectively. Association analysis showed that the backfat thickness of the AA and AC genotypes was significantly higher than that of the CC genotype (AA *vs* CC, $p = 0.0041$; AC *vs* CC $p = 0.0019$). No significant differences were observed between the three genotypes for age up to 70 kg (Table 2).

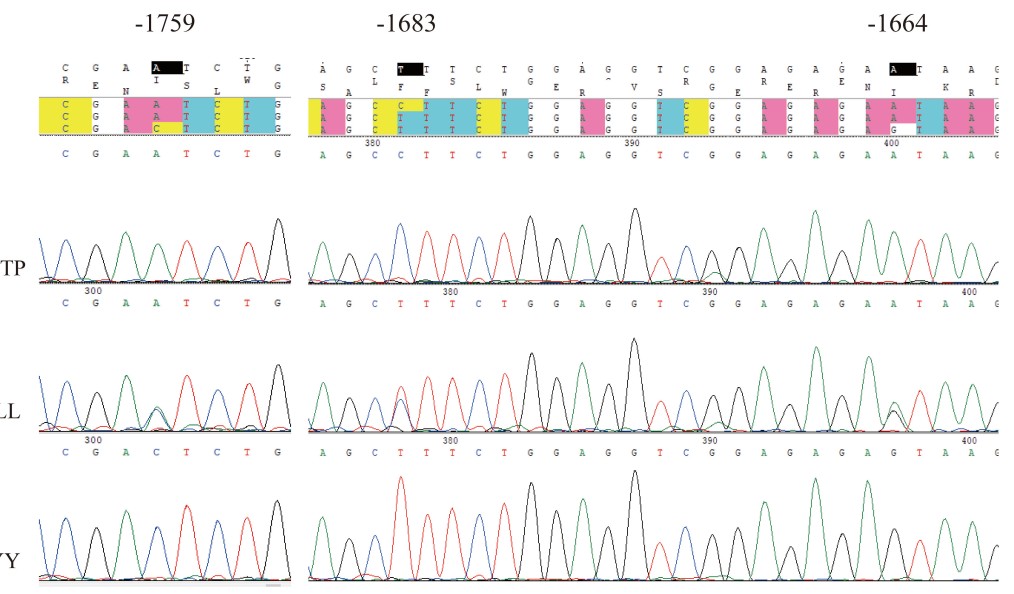

**Figure 3** Sequencing map of three linked sites. TP, Tibetan pig; YY, Yorkshire; LL, Landrace.

**Table 1 Gene and genotype frequencies of A-1759C sites in different pig breeds.**

| Breed | Genotype frequency (number/frequency) | | | | Allele frequency | |
|---|---|---|---|---|---|---|
| | **AA** | **AC** | **CC** | **HWE $\chi^2$ value ($P$-value)** | **A** | **C** |
| TP | 32/0.9412 | 2/0.0588 | 0/0.0000 | 0.031 ($P = 0.860$) | 0.9706 | 0.0294 |
| LL | 2/0.0588 | 12/0.3529 | 20/0.5882 | 0.013 ($P = 0.911$) | 0.2353 | 0.7647 |
| YY | 6/0.15 | 17/0.425 | 17/0.425 | 0.137 ($P = 0.934$) | 0.3625 | 0.6375 |

**Table 2 Association analysis of A-1759C with back fat thickness and growth traits.**

| Genotype | AA ($n = 34$) | AC ($n = 58$) | CC ($n = 12$) |
|---|---|---|---|
| Thickness of back fat | $14.16 \pm 3.17$[a] | $14.24 \pm 3.28$[a] | $10.95 \pm 2.11$[c] |
| Day old at 70 kg | $216.36 \pm 15$ | $213.24 \pm 16.11$ | $214.14 \pm 12.32$ |

**Notes.**
[a,c]Lowercase letters following the data indicate significant differences. Interphase letters indicate significant differences ($P < 0.01$); the same letters indicate non-significant differences ($P > 0.05$).

## Transcription activity of *AACS* promoter region

Four length fragments of the 5′ flanking region (−2044/+116, −1493/+116, −1050/+116, and −608/+116) were amplified, and the promoter activity in these four regions was detected using the dual-luciferase reporter assay. The results indicated that the fluorescence activity of these four sections was significantly higher than that of the control group (PGL3-control), indicating that these four sections contained critical regulatory elements. Among these, the −608/+116 region may have vital regulatory elements that enhance promoter activity, and the −1493/−2044 region may have critical active elements

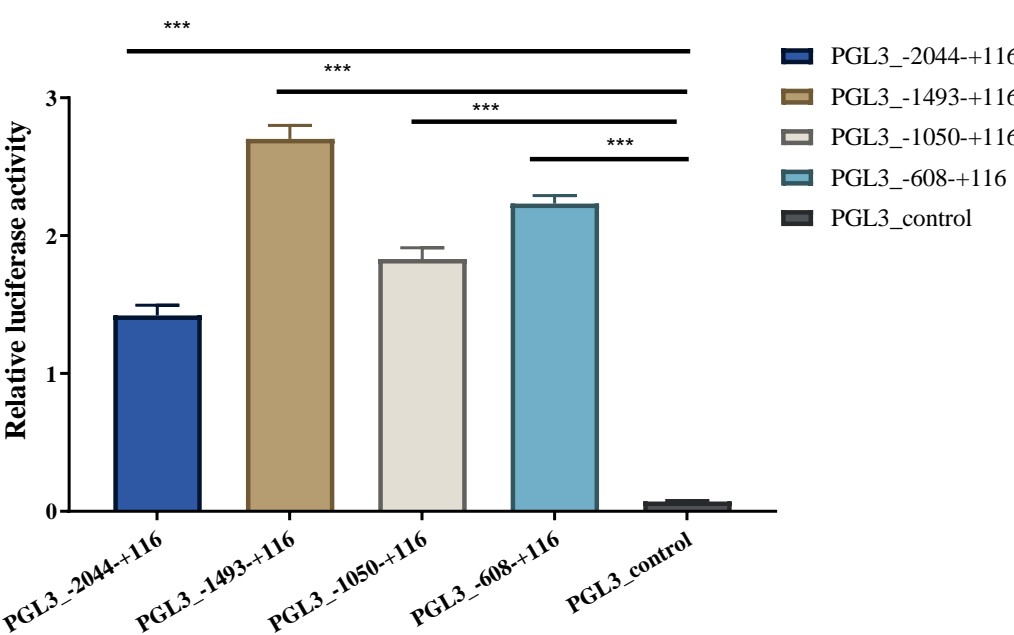

**Figure 4  Dual-luciferase analysis for promoter activity.** The control group was co-transfected with PGL3-basic (900 ng) and PRL-TK (100 ng). The target plasmid (PGL3_−2044–+116, PGL3_ −1493–+116, PGL3_ −1050–+116, PGL3_ −608–+116; 900 ng/each) was co-transfected with the internal control plasmid pRL-TK (100 ng). There were 4 replicates in each group. Each bar rep-resents the means ± SD. ***$p < 0.001$.

that inhibit promoter activity (Fig. 4). The identified A-1759C, C-1683T, and A-1664G linkage sites were all located in the −1493/−2044 region; thus, their polymorphisms might affect the expression of *AACS* and fat deposition in pigs.

### *AACS* inhibits the proliferation of subcutaneous preadipocytes

Porcine subcutaneous preadipocytes were infected with a lentivirus to overexpress *AACS*, and unsuccessfully infected cells were removed using puromycin. The results confirmed the successful overexpression of *AACS* using lentiviral transfection (Fig. 5A). The CCK8 assay demonstrated that the number of living cells decreased significantly in the *AACS* overexpression group compared with that in the control group (Fig. 5B). Similar results were obtained in the EDU assay. *AACS* overexpression resulted in reduced EdU positivity compared to the control group (Fig. 5D). We detected the mRNA expression levels of CDK4 (cyclin-dependent kinase 4) and cyclin B, which are considered proliferation-marker genes, and found that the expression levels of both genes decreased after *AACS* overexpression, with cyclin B showed significant differences (Fig. 5C).

### *AACS* inhibits adipogenic differentiation of subcutaneous preadipocytes

To validate the *AACS*-mediated regulation of the differentiation of subcutaneous preadipocytes, Oil Red O staining was performed on day 6 of preadipocyte differentiation to detect the number of lipid droplets generated (Fig. 6A). The results revealed that *AACS*

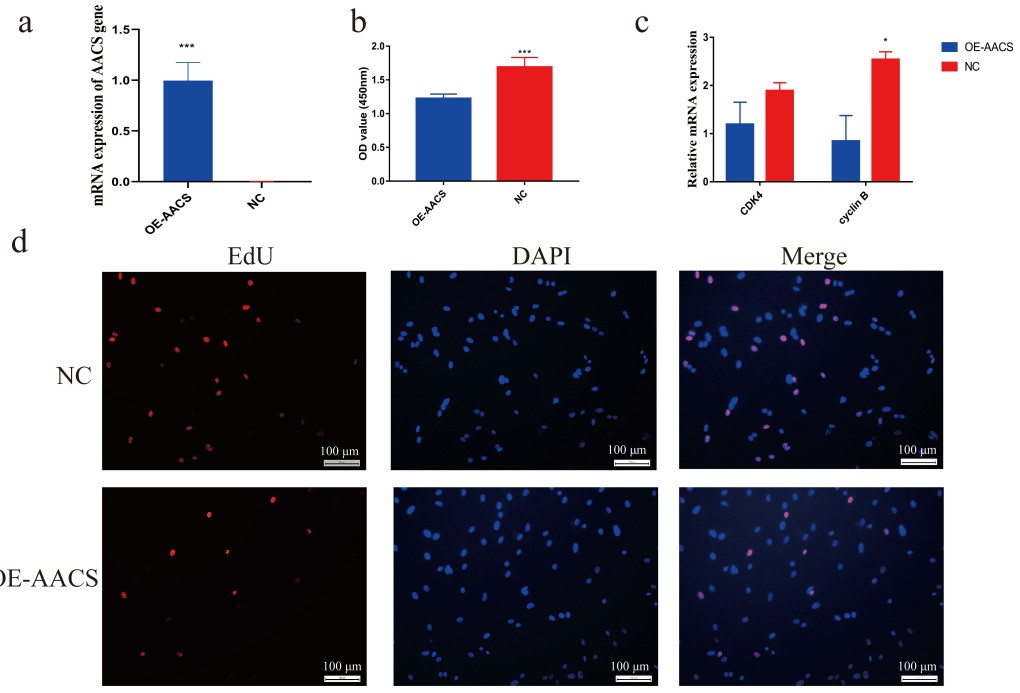

**Figure 5  Overexpression of *AACS* inhibits the proliferation of porcine subcutaneous preadipocytes.**
(A) Detection of overexpression efficiency after lentivirus infection. (B) Overexpression of *AACS* inhibited the proliferation of porcine subcutaneous preadipocytes using CCK8 assay ($n = 9$). (C) The mRNA expression levels of proliferation-related genes ($n = 3$). (D) The proliferation of porcine subcutaneous preadipocytes after overexpression of *AACS* for 24 h was detected using EdU staining ($n = 3$). Each bar represents the means ± SD. *$P < 0.05$, ***$P < 0.001$.

overexpression in porcine preadipocytes significantly reduced the lipid droplet production (Fig. 6B), and the expression of the marker genes of adipogenesis, *PPAR γ*, *CEBP α*, *AP2*, and *SREBP-1C*, were significantly lower in the overexpression group than in the control group (Fig. 6C). These results indicate that *AACS* plays an inhibitory role in the differentiation of subcutaneous preadipocytes in pigs.

## DISCUSSION

This study is the first to clone *AACS* mRNA sequence and report that *AACS* plays an inhibitory role in the differentiation of porcine subcutaneous preadipocytes. These findings confirm the crucial role of *AACS* in porcine subcutaneous adipogenesis and proliferation. Our findings also elucidate the potential effect of *AACS* on the improvement of subcutaneous fat deposition and provide theoretical implications for its role in the improved breeding of pigs.

 *AACS* directly activates ketone bodies in the cytosol to synthesize cholesterol and fatty acids (*Endemann et al., 1982*). We observed *AACS* expression in the adipose tissue of DY pigs with divergent backfat thicknesses, suggesting that *AACS* plays a role in the subcutaneous fat deposition in pigs. Porcine *AACS* genes were cloned and compared using

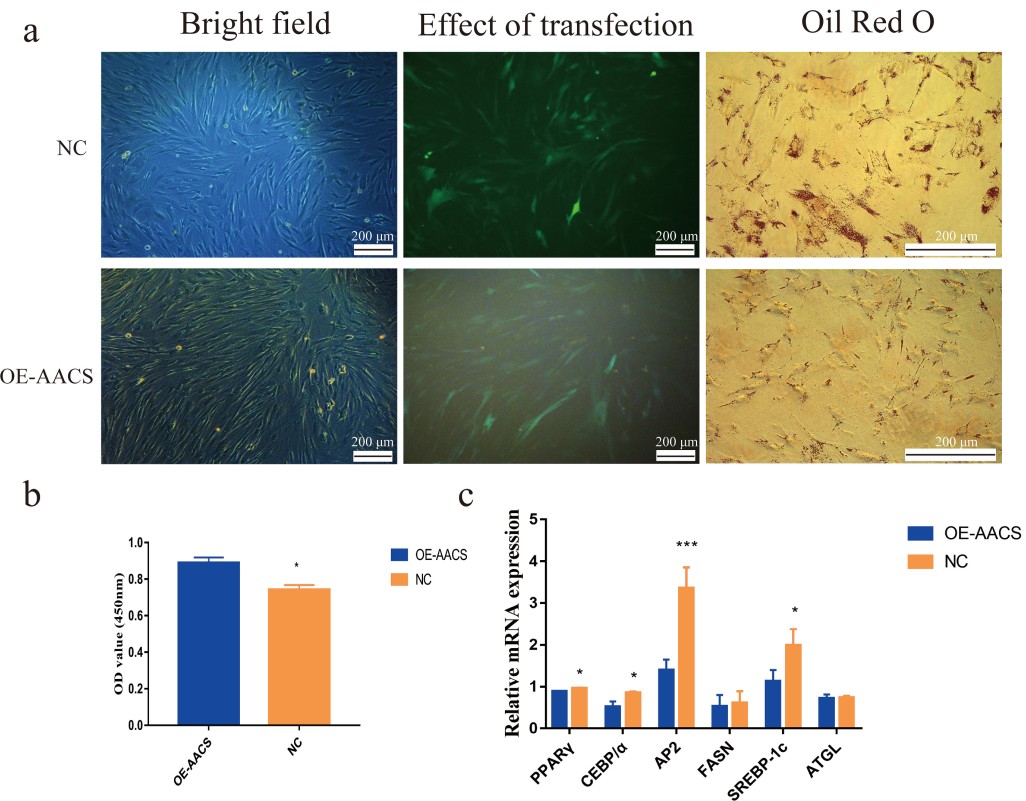

**Figure 6   Overexpression of *AACS* inhibited the differentiation of porcine subcutaneous preadipocytes.** (A) Image of porcine adipocytes stained with oil red O after six days of differentiation. (B) Oil-red O staining was used to determine the content of lipid droplets in porcine subcutaneous adipocytes ($n = 3$). (C) The mRNA expression of marker genes associated with adipogenesis in porcine subcutaneous adipocytes ($n = 3$). Each bar represents the means ± SD. *$P < 0.05$, ***$P < 0.001$.

the NCBI for Biotechnology Information website. The high similarity of amino acid and nucleotide sequences across species indicates that the mRNA sequence of the cloned AACS gene is relatively accurate and that the gene is relatively conserved among species.

Highly conserved genes often share common regulatory mechanisms (*Chikina & Troyanskaya, 2011*; *Gerstein et al., 2014*; *Stuart et al., 2003*). This study, along with others, demonstrated that *AACS* genes are highly expressed in adipose tissue (*Yamasaki et al., 2007*) and play a negative regulatory role in the deposition of subcutaneous fat in pigs and that upregulation of *AACS* inhibits the differentiation of subcutaneous preadipocytes. In contrast, in 3T3-L1 cells, the downregulation of *AACS* inhibited cell differentiation (*Hasegawa et al., 2012*). Similar results have been found in mice, where the *AACS* gene expression level in white adipose tissue was lower in Zucker fatty rats than in lean rats. However, in high-fat, diet-induced obese rats, the expression levels of *AACS* were increased (*Yamasaki et al., 2007*). In the epididymal adipose tissue, *AACS* protein expression decreases in mice fed a short-term high-fat diet (*Plubell et al., 2017*). The expression trend of *AACS* in different species indicates that this is not only affected by obesity and species type but also

by the location of fat deposition, where the mechanism of fat deposition differs at different locations (*Luo et al., 2022*; *Mendizabal et al., 1997*; *Zhou et al., 2010*). The formation of adipose tissue mainly involves an increase in the number of adipocytes (proliferation) and their hypertrophy (differentiation) (*Choe et al., 2016*; *Ghaben & Scherer, 2019*). High expression of *AACS* inhibits the proliferation and differentiation of porcine subcutaneous preadipocytes, resulting in a decrease in flat droplet generation.

Active promoter elements with vital functions are present in the 5′ flanking region of the gene (*Kimura et al., 2001*; *Li et al., 2017*). We detected the promoter activity of this region and observed that each flanker region gained strong promoter activity by truncating the amplified fragment, among which the −1,493/−2,044 region had important promoter-inhibiting active elements. Notably, an important linkage mutation site (A-1759C) was identified in this region, validated in a population of 104 pigs, and demonstrated to be significantly associated with backfat thickness, which may be a vital molecular marker for pig breeding.

# CONCLUSIONS

We cloned porcine *AACS* mRNA sequences for the first time and confirmed that *AACS* is a negative regulator of subcutaneous preadipocyte differentiation in pigs. This study provides evidence that elevated *AACS* genes lead to the downregulation of PPAR $\gamma$, CEBP/$\alpha$, and AP2 genes which are key regulators of fat formation. Additionally, we explored luciferase activity in the 5′ flanking region of the *AACS* gene and obtained molecular markers that could be used for genetic improvement in pigs. Although we present preliminary findings, we confirmed that *AACS* is a vital regulator of porcine subcutaneous fat deposition and can have practical implications in improving pig breeding.

## Funding
This work was supported by the Tibet Major Science and Technology Project (XZ202101ZD0005N) and the National Natural Science Foundation of China (32060736). The funders had no role in study design, data collection and analysis, decision to publish, or preparation of the manuscript.

## Grant Disclosures
The following grant information was disclosed by the authors:
Tibet Major Science and Technology Project: XZ202101ZD0005N.
National Natural Science Foundation of China: 32060736.

## Competing Interests
The authors declare there are no competing interests.

## Author Contributions
- Pan Zhang conceived and designed the experiments, performed the experiments, analyzed the data, prepared figures and/or tables, and approved the final draft.

- Bo Zhang conceived and designed the experiments, analyzed the data, prepared figures and/or tables, authored or reviewed drafts of the article, and approved the final draft.
- Yu Fu performed the experiments, prepared figures and/or tables, and approved the final draft.
- Pan Li performed the experiments, prepared figures and/or tables, and approved the final draft.
- Hao Zhang conceived and designed the experiments, authored or reviewed drafts of the article, and approved the final draft.

## Animal Ethics

The following information was supplied relating to ethical approvals (*i.e.*, approving body and any reference numbers):

All experiments were approved by the Committee on the Ethics of Animal Experiments of China Agricultural University (SKLAB-2012-04-07).

## Data Availability

The full mRNA sequences of AACS are available at GenBank: OP807955.

The raw data is available in the Supplemental Files.

## Supplemental Information

Supplemental information for this article can be found online at http://dx.doi.org/10.7717/peerj.16406#supplemental-information.

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
