# Peer review of "Cloning and functional characterization of porcine AACS revealing the regulative roles for fat deposition in pigs"

_PeerJ, doi:10.7717/peerj.16406_

## Round 0.1 · original submission · Major Revisions

Dear Dr. Zhang,

Thanks for your submission to PeerJ.

Three reviewers reviewed your manuscript. While all reviewers are positive about the merit of the manuscript, they also indicated that there are major issues that need to improve before your manuscript be received for publication.

·

Basic reporting

This study centered the AACS gene, cloned the full-length sequence of the gene, analyzed the sequence features, identified the SNPs differences in the 5' upstream region of the AACS gene and its correlation with porcine back fat thickness; and finally detected the effect of the AACS gene on the differentiation of porcine adipocytes by overexpressing AACS. The experimental design is good, but his manuscript is still flawed and needs to be checked for revisions before acceptance. Comments are as follows:

Major comments:
(1) In the Introduction or Method section, it is important to clarify which of the pig breeds selected for this study are high or low back fat thickness pigs, respectively. Meanwhile, in ‘Introduction’ (L33-34), there are many ways to reduce subcutaneous fat, so why must we study candidate genes, transition logic is missing here.
(2) Why only SNPs during 5’-flanking region of AACS were screened, not CDS region of AACS, or not whole gene? The sound logic or reasons for doing so should be outlined in the background. (2) For SNPs screening and genotyping, how did you genotype them, genotyping with a DNA pool of the same pig breed or separate DNA sample of each individual, all these should be covered in Introduction or Method section.
(3)qPT-PCR, the amplification cycle system needs to be provided (as supplementary file).
(4) Figures: Fig4, what does y-axis represent? Fig5b, 5c, the y-axis of both plots is not clear enough and needs to be redrawn.
(5) The English grammar of whole text needs further improvement.


Minor comments:
Line 79: ‘20 µL reaction volume were reverse transcribed to cDNA’, Reaction systems and reaction procedures for reverse transcription cDNA should be provided.
L95: ‘quantitative real-time PCR (qPCR)’. Here it’s qPCR, however, in Fig2, it was written as SqRT-PCR. If it is different from qRT-PCR, please check and keep consistent through the text.
L170: For signal peptide result, the method used to predict the signal peptide should be provided in Method section.
L171-172: nucleotide similarities and amino acid sequence similarities, the methods or websites used for analyzing these should be provided in Method section.
L180: For abbrev BF, please give the full name of BF at the first time.
L181-182: ‘lower in TP than in YY tissues, and no significant differences were observed between these in LD tissue’. The AACS expression was examined in two tissues of two pig breeds, TP and YY are names of pig breeds. So, the tissues should be changed to breeds. Moreover, this sentence should be revised and improved for better understanding.
L182-183: ‘Similar results were obtained in DY pigs with divergent back fat thicknesses’. This manuscript didn’t show any Figures or Supple Figures about this result.
L183-185: The audience doesn’t know which pigs belong to ‘high back fat thickness (HBF) group’ or ‘low back fat thickness (LBF) group’, all these should be addressed clearly in Method section (Experimental materials).
L190: ‘Thus, the frequencies of allies and genotypes of A-1759C could represent the polymorphisms at these three sites’. (1) How this conclusion was reached? (2) ‘allies’, do you mean ‘alleles’?
Table1: The method of calculating HWE should be provided in Method section.
Reference: The reference ordering (the sixth reference has a number in front of citation) needs to be adjusted.
Supplementary uncropped gels: The marker and band size of product were missed in all supplementary figures.
There is no ‘Conclusion’ section in this manuscript. Please check and make sure this is consistent with the journal formatting requirements.

Experimental design

no comment

Validity of the findings

no comment

Additional comments

no comment

Reviewer 2 ·

Basic reporting

This article has done a comprehensive analysis to reveal the importance of AACS gene in adipocyte differentiation. The English is clear and unambiguous but still has room to improve. While most of the figures and tables are good, some of them lack legends and do not appear in their original scale.

Experimental design

Research question is well defined. Through overexpression of AACS of the porcine subcutaneous preadipocytes, the authors revealed that AACS inhibits the proliferation and adipose differentiation of subcutaneous preadipocytes. However, it's needed to indicate the replicate number of the experiments which is quite essential to check the robustness of the results. By investigating the allele frequency of variants located in the 5'flanking regions and the promoter activity, the authors demonstrate the potential role of the variants as the markers. I cannot see a direct relationship between the three variants and the promoter activity. It's ok that the variants potentially affect the back fat thickness, but no direct evidence that the variants will influence the gene expression of AACS and thus induce the changes of the promoter activity. Therefore, I think this part is less convincing.

Validity of the findings

no comment

Additional comments

1. Title is not appropriate. It's like a title of proposal. I would expect some title like "cloning and functional characterization of porcine AACS revealing the XXXX" to conclude the main contribution of the paper fully.

2. Abstract: 23-24 Is one of the three SNPs within the active regulatory elements? Please rewrite these sentences to make them more connective.

3. L49: identified
4. L180: You should spell out the full term at its first mention.
5. L181: I think TP and YY are breeds? Suggest “tissues” change to “pigs”.
6. L183: Since you’ve already defined “BF” as the abbreviation of back fat… “In back fat tissue” should be “In BF tissue”. Please check all over the manuscript to keep it consistent.

7. L185: You should include the replicate numbers in each group in Figure 2b and Figure 2C (also Figure 4, Figure 5, Figure 6). I think you only have 6 pigs in DY breed. If only two or only one pig in the low back fat thickness group, it could be an outlier. It seems you only have one sample in the low back fat thickness group, since I cannot see any error bar in this group in Figure 2C. I suggest more animals be included in this group to improve the confidence of the result. Furthermore, the manuscript never mentions the criterion of discrimination of high/low back fat thickness.

8. L187: For this part, have you observed the expression differences of the different alleles?

9. L189: Figure 3 is uninformative, even the alleles are missing in the sequencing map of YY pig. Did only these three SNPs found after cloning and sequencing?

10. L201: You should indicate what the numbers mean in table 2.
11. 241: “between species” to “across species”

·

Basic reporting

In this paper, the authors cloned the full length of the porcine AACS mRNA sequence. They identified three fully-linked SNP sites in the AACS promoter region, A-1759C, 22 C-1683T, and A-1664G, that may be vital molecular markers for regulating back fat thickness. Using overexpression experiments, they observed that AACS inhibited the proliferation and differentiation of porcine subcutaneous preadipocytes. Overall, the paper is generally well-written and structured.

Experimental design

In my opinion, the methods are not described in sufficient detail that the experiments can be reproduced by others.
Specific points to be addressed:
1. It is necessary to provide a detailed description of the SNPs identification method, such as whether DNA pool sequencing or single sample sequencing is used, and the sample size for sequencing?
2. Which internal control vector is used for the dual luciferase assay? What is the transfection ratio of the target vector and internal control? How is it determined?
3. How is the promoter sequence determined? Why is the -2044 - +116 fragment amplified? Have you used bioinformatics analysis to predict porcine AACS promoter regions?
4. Please supplement the method used for the induction of porcine preadipocyte differentiation.
5. It is recommended to analyze whether the identified SNPs alter transcription factor binding.
6. The legend of Figure 4 needs to be more detailed, such as the amount of vector transfection, the number of repetitions, etc.
7. Please check whether the data is displayed as mean ± SD.

Validity of the findings

No comments

Additional comments

No comments

---

## Round 0.2 · Minor Revisions

Dear Dr. Zhang,

Thank you very much for your submission to Peer J. Your manuscript "Cloning and functional characterization of porcine AACS revealing the regulative roles for fat deposition in pigs" has been reviewed and there are some minor points that may need your attention. Please revise your manuscript accordingly and I am looking forward to your revised manuscript.

·

Basic reporting

Authors have addressed all comments clearly. Now the manuscritp looks nice for acceptance.

Experimental design

no comment

Validity of the findings

no comment

Additional comments

no comment

Reviewer 2 ·

Basic reporting

The manuscript has been largely improved and the authors have addressed most of my concerns. But I still have some minor comments.

Minor comments:
1. L22: “highly expressed in subcutaneous fat” to “expressed highest in subcutaneous fat”.
2. L67: what's the relationship between Berkshine Dingyuan Pig and Dingyuan Pig? Is Berkshine Dingyuan pig the hybrid breed of Berkshine and Dingyuan? Please clarify it. I’ve never heard of such a name of breed. If so, you’d better tell where DY breed comes from in L72.
3. L205: Why are there no direct measurement values (CT values) for the gene expression of each tissue?
4. L226: better to show the exact P-values.
5. Figure 3: The genotypes for YY breed are still missing.
6. Figure 5 & Figure 6: Please provide the number of replicates.

Experimental design

no comment

Validity of the findings

no comment

·

Basic reporting

The authors have addressed all my concerns. I consider the manuscript acceptable for publication.

Experimental design

no comment

Validity of the findings

no comment

---

## Round 0.3 · accepted · Accept

Dear Dr. Zhang,

Thank you very much for the submission and revisions.